# Serial Monopoly on Blockchains with Quasi-patient Users

Paolo Penna[*]                Manvir Schneider[†]

IOG                       Cardano Foundation

May 2024

**Abstract**

This paper introduces and investigates an extension of the price dynamics in serial monopoly blockchain described in Nisan [Nis23], tailored to accommodate *quasi-patient* users. Our model reflects users' diminishing interest in having their transactions added to the ledger over time, resulting in only a fraction $\delta$ of the current demand persisting in the subsequent round. The framework presented by Lavi et al. [LSZ22], where users are impatient and derive utility only from immediate transaction inclusion in the next block, corresponds to $\delta = 0$. Fully patient users who wait forever as in [Nis23], correspond to $\delta = 1$ in our model. This work provides new bounds on the price dynamics for the more interesting case $\delta \in (0, 1)$, showing somewhat unexpected effects on the dynamics itself. While the dynamics for the fully patient case is essentially "oblivious" of the structure of the daily demand curve, this is no longer true for finite $\delta < 1$. Moreover, the dynamics undergoes a "transition phase" where for some $\delta$ it behaves as in the fully patient setting ($\delta = 1$), and for some smaller values $\delta' < \delta$ it stops "oscillating" and stays at the highest ("monopolist") price. We provide quantitative bounds and analytical results that apply to different demand functions showing that the bounds for $\delta = 1$ are not tight in general, for $\delta < 1$. These provide guarantees on the minimum ("admission") price such that transaction willing to pay that price are eventually included (and those who do not want are never included).

## 1   Introduction

Transaction fee mechanisms are a fundamental part of a blockchain. A block leader, in general, has full freedom to choose which transaction from the mempool (and private transaction pool) to include in a block. A well designed transaction fee mechanism contributes to maximizing the social welfare, which

---

[*]paolo.penna@iohk.io

[†]manvir.schneider@cardanofoundation.org

is the total value of the chosen transactions subject to the block size constraint. The most prominent proof-of-work blockchain, Bitcoin, employs a pay-your-bid mechanism. In particular, the higher the bid attached to a transaction, the higher the chances to be included in the next block by the miner. The user-paid bids are rewarded to the miner. Rational miners will always try to maximize their revenue and will therefore choose the highest paying transactions. Since block space is scarce, this mechanism can drive prices high, especially when there is congestion and hence transaction inclusion might experience a considerable delay. A recent example is the Bitcoin halving at block 840,000. Users paid insane amounts in fees for their transaction to be included. For example one user paid 800M Sats (approx. $500K) in transaction fee for a transaction transferring $0.70. In total block 840,000 generated $2.4M in fees.[1] Many users wanted their transaction to be included in this historical block and were willing to pay huge amounts of transaction fee. This came at the cost that "normal" users had to experience a bigger delay for their transactions to be included.

Opposing to the pay-your-bid mechanism is the dynamic posted price mechanism, like EIP-1559. In Ethereum's EIP-1559, the transaction fee is split into base fee and a tip. The block proposer only receives the tips while the base fee is burned. Rational block proposers will therefore only select transactions that pay at least the base fee.[2] The target of EIP-1559 is to have half-full blocks. If the previous block was filled less (more) than the target, the base fee is lowered (increased) accordingly. The dynamic base fee and the target of half-full blocks allows to handle low and high demand phases.

A different approach to transaction fee is that of Cardano where there is a fixed fee (per byte). The transaction selection process involves a first-in-first-out (FIFO) mechanism. The transaction fees of included transactions are collected in a pool and distributed at the end of an epoch (5 days). The distribution ensues proportional to the proposed blocks in the epoch. High priority transactions willing to pay higher fees may not experience faster transaction inclusion.[3]

A natural question to ask is whether block leaders should be allowed to set their own fees rather than a fee imposed by the protocol. The study of such a monopolist pricing mechanism is part of Nisan [Nis23] and this paper. In the monopolist pricing mechanism, transactions willing to pay at least the price set by the monopolist are included in a block (until the block is full). Unlike in the pay-your-bid mechanism, all included transactions pay exactly the price set by the monopolist (rather than their bid). Or, in the words of Lavi et al.[LSZ22], the monopolist chooses the number of accepted transactions in the block and all transactions pay the smallest bid among the accepted transactions.

In his paper [Nis23], Nisan assumes that block leaders set their own prices

---

[1]See here: https://www.blockchain.com/explorer/blocks/btc/840000 (accessed on April 25, 2024)

[2]It may be that a block proposer has some positive intrinsic value for some transaction that pays less than the base fee, and therefore includes this transaction by paying the remaining base fee himself. Note that this is related to active block proposers, see Bahrani et al. [BGR24].

[3]There is a proposal to split the blocks in different tiers with different fees. Priority tiers have higher fees and ensure that high priority transactions paying enough fees get fast inclusion in the eligible tier, see Kiayias et al. [KKLP23].

and all transaction that are not included in a block remain in the mempool forever until they are picked up eventually in a future block. This assumption, however is too strong and does not reflect real world behavior. Impatient users may cancel their transaction after some time if not included in a block. We assume that only a fraction of unsupplied transactions remain in the next round.

## 1.1 Our Contributions

We put forward a model for monopolist pricing dynamics tailored to accommodate *quasi-patient* users (see Section 2 for details and formal definitions). Our model incorporates a "decay" parameter $\delta \in [0, 1]$ which corresponds to the fraction of pending transactions that remain in the mempool at the next round (thus, a fraction $1 - \delta$ of pending transactions gets withdrawn by the users at each round). The case of *impatient users* in Lavi et al. [LSZ22] corresponds to $\delta = 0$, and the case of *patient users* in Nisan [Nis23] to $\delta = 1$. Our model spans all intermediate scenarios between these two extreme cases, and it allows us to study how monopolist pricing mechanisms behave at different *patience* levels $\delta$. In particular, we no longer assume *patient users* who are willing to wait (and pay) indefinitely long for their transactions to be included on the ledger.

We provide analytical results on the monopolistic pricing dynamics for any $\delta \in (0, 1)$. Our findings highlight that monopolistic price mechanisms for *quasi-patient* users still posses good features, though with some key differences with the case of *patient* users. In Section 3, we analyze the dynamics for different values of $\delta$ and how it affects their behavior. In particular, we demonstrate that in regimes with a *sufficiently small fraction of expiring transactions* ($\delta < 1$ sufficiently large), the dynamics behaves qualitatively similarly to the case of *no expiring transactions* ($\delta = 1$). The analysis highlights several differences, particularly how the "structure" of $Q$ influences the dynamics for $\delta < 1$ compared to the case $\delta = 1$.

Specifically, Theorem 2 informally states that:

- *Prices decrease or jump up to maximum price.* The price for being included in the current block is either smaller than the one at the previous block, or it is the maximum price.

- *Immediate inclusion price (monopolistic price).* This is the largest price that the monopolist ever asks, which is also the price that is always asked if users are *impatient* and there is no pent-up demand [LSZ22].

- *Minimum admission price.* Pent-up demand due to *quasi-patient users*, makes the price fluctuate over time between the monopolistic (maximum) price and some *minimum admission price*.

A direct comparison between our bounds for quasi-patient users and the case of patient users, shows the following. First, our upper bounds on the minimum admission price depend on $\delta$. Second, the minimum admission price for $\delta < 1$ is at least the minimum admission price for $\delta = 1$ (cf. our Theorem 2 and

Theorem 1 below from [Nis23]). In Section 3.2, we prove lower bounds on the minimum admission price. In particular, Theorem 4 states the following:

- *The minimum admission price for impatient users is never tight for quasi-patient users.* That is, for every $\delta < 1$ there is a demand function for which the minimum admission price is strictly higher.

- *Collapse to the impatient case.* The change in the dynamics is not continuous in $\delta$. For some small positive $\delta > 0$, the dynamics behave exactly like the *impatient users* case ($\delta = 0$). That is, the minimum admission price coincides with the (maximum) monopolist price at all time steps.

Furthermore, the above mentioned collapse means that the positive effect of pent-up demand, which results in a minimum admission price smaller than the monopolist price, may completely be nullified for quasi-patient users. Another important difference is that, for patient users, the minimum admission price is "almost" independent of the structure of the demand function (it only depends on $s$ and on the revenue at the monopolist price). This is no longer true for quasi-patient users, where the "overall strcture" of the demand function seems to play a role.

## 1.2 Related Work

Transaction fee mechanisms are analyzed from the perspective of mechanism design in Roughgarden [Rou21]. Additionally the dynamic posted price mechanism EIP-1559 is analyzed. Follow up work on transaction fee mechanism design includes [FMPS21, CS23, CRS24, CSLZZ24, BGR24]. More work focusing on the dynamics of EIP-1559 is [LMR+21, RSM+21, LRMP23].

**Monopolistic pricing mechanisms.** Monopolistic pricing mechanisms have been studied in Lavi et al. [LSZ22], Yao [Yao18] and Basu et al. [BEOS19] prior to Nisan [Nis23] and this paper. Lavi et al. [LSZ22] study the monopolistic pricing mechanism and describe the mechanism as follows: (1) Transactions specify bids (maximal fee) they are willing to pay; (2) Miners (or block leaders/monopolists) choose which subset to include in their block; (3) All transactions in the block pay the exact same fee which is equal to the smallest bid among the included transactions; (4) Miners maximize their revenue which is the product of the minimal bid and the number of included transactions. The focus of their paper is on a single shot game where users are maximally impatient in the sense that they derive utility from immediate block inclusion and no utility for inclusion in a future block. They show that truthful bidding (users bidding their true valuation) is "nearly" an equilibrium, i.e. relative gains from strategic bidding go to zero as number of transactions increase. The revenue achieved monopolistic pricing mechanism collects at least as much revenue from maximally impatient users as the pay-your-bid mechanism (as employed in Bitcoin). Yao [Yao18] builds on the work of [LSZ22] and studies properties of the monopolistic pricing mechanism, in particular, incentive compatibility when user's

valuations are drawn from an i.i.d. distribution. Basu et al. [BEOS19] study a similar setting to the one of [LSZ22], however, in their model, they consider many miners with the goal of maximizing social welfare. Note that the model of [LSZ22] does not aim to maximize social welfare. To see this, note that, if the monopolist chooses a subset of transactions that does not fill the block entirely, the monopolist could potentially include transactions with lower bids. However, doing so would decrease the price that all included transactions have to pay and hence would lower the monopolist's revenue.

Nisan [Nis23] studies a monopolist pricing mechanism, in which each block leader (or proposer) is allowed to choose the price $p$ for his block. Transactions willing to pay at least $p$ may be included by the monopolist and all included transactions pay exactly $p$.[4] Rationality of block leaders implies that the block leaders will choose a price that maximizes their revenue given price and the block space filled by the chosen transactions.[5] Nisan's model involves infinitely patient users, i.e. users' valuations of transaction inclusion do not depreciate over time. Transactions stay in the mempool until eventually picked up by some block leader. Block leaders face the same demand distribution at every step in time plus the pent-up demand from the previous steps, that is, additionally to the daily demand the block leaders faces the transactions that were not picked up by previous block leaders. When optimizing given the current total demand, the block leader only optimizes for the current block (myopic block leader). Furthermore, the available block space for each block is fixed and demand is known to the block leader.

Kiayias et al. [KKLP23] study a mechanism to account for transactions with different priority/urgency. In particular, the mechanism splits blocks into different tiers with each tear having its own characteristics such as fee and size. The fee and size are dynamically adjusted based on previous demand and fees. This mechanism ensures that high priority transactions can choose to be included in a tier with high priority by paying high transaction fee.

## 2    Model

We extend the model of Nisan [Nis23] for non-strategic agents with an additional parameter $\delta \in [0, 1]$ which corresponds to the fraction of pending transactions remaining at next round (thus, $1 - \delta$ is the fraction of pending transactions withdrawn by the corresponding users – see below). The dynamics is specified as follows:

- *Time* is discrete and indexed by $t = 1, 2, \ldots,$.

---

[4]In principle, this mechanism is the same as in Lavi et al. [LSZ22]. In [LSZ22] the fee to be paid by users is determined by the lowest bid $p$ of the included transactions. There is at least one transaction with bid $p$ (which is the lowest bidding transaction), while in Nisan [Nis23] there need not be a transaction with bid exactly equal to $p$.

[5]While the other mechanism of Bitcoin and Ethereum maximize the total value of included transactions subject to the available block space (i.e. social welfare), the monopolist mechanism maximizes the block leaders revenue.

- *Daily demand*: A demand function $Q$ quantifies the daily demand $Q(p)$ for every price level $p$. Function $Q$ is continuous and decreasing in $p$ as $Q(p)$ is the number of newly added transactions willing to pay $p$ or more to be included.

- *Monopolist*: A monopolist (chosen for the current round $t$) faces a total demand $D_t$ consisting of daily demand and pent-up demand from previous rounds (see below). As $D_t(p)$ is the total number of transactions willing to pay at least $p$, the monopolist chooses a price maximizing his own revenue subject to the supply constraint $s$ (block size = max number of transactions per block):

$$p_t = \arg\max_p p \cdot \min(s, D_t(p)) \ . \tag{1}$$

  The corresponding supplied quantity is $q_t = D_t(p_t)$, and the monopolist's revenue (at time $t$) is $\mathtt{REV}_t := p_t \cdot q_t$.

- *Pent-up demand*: Initially there is no pent-up demand from previous rounds, that is, $Z_0(p) = 0$ for all $p$. The pent-up demand at time $t \geq 1$ is

$$Z_t(p) := \begin{cases} D_t(p) - q_t & \text{for } p \leq p_t \\ 0 & \text{for } p > p_t \end{cases} \ . \tag{2}$$

- *Total demand and $\delta$*: Only a fraction $\delta \in [0,1]$ of pent-up demand survives to the next round, and thus total demand is

$$D_t(p) = \delta \cdot Z_{t-1}(p) + Q(p) \ . \tag{3}$$

**Remark 1.** *For $\delta = 1$ the model above boils down to the one in [Nis23] where all transactions not included in the current round remain in the system and they are eventually included if an only if their price is above some minimum price $p_{ser}$. For $\delta = 0$ transactions are either immediately included or they disappear, thus implying that the dynamics above stay at the monopolist price $p_{mon} > p_{ser}$ and only transactions willing to pay this price are included.*

In the sequel we shall focus on the case $\delta \in (0,1)$ as the case $\delta = 0$ is trivial and $\delta = 1$ coincides with the model in [Nis23].

**Key quantities.** Note that by the definition of the total demand and pent-up demand we can write the total demand as follows.

**Remark 2.** *The total demand at time $t$ can be rewritten as*

$$D_t(p) = a_t \cdot Q(p) - b_t \tag{4}$$

*where from* (2) *and* (3) *we have*

$$a_t = 1 + \delta + \cdots + \delta^{t-1} \ , \qquad b_t = q_1 \delta^{t-1} + q_2 \delta^{t-2} + \cdots + q_{t-1} \delta \ . \tag{5}$$

*Note that $a_t = \frac{1-\delta^t}{1-\delta}$ for $\delta \in (0,1)$, and $a_t = t$ for $\delta = 1$.*

As our main results (see Section 3) show, the price dynamics fluctuate between two prices that involve the following quantities:

**Definition 1.** For any demand function $Q$ and any supply $s$, the corresponding *monopolist price* $p_{mon}$ and *serial price* $p_{ser}$ are defined as follows:

$$p_{mon} := \arg\max_p p \cdot Q(p) \ , \qquad\qquad q_{mon} := Q(p_{mon}) \ , \qquad (6)$$

$$p_{ser} := p_{mon} \cdot q_{mon}/s \ , \qquad\qquad q_{ser} := Q(p_{ser}) \ . \qquad (7)$$

Note that the *monopolist price* $p_{mon}$ is simply the price that maximizes the revenue of the monopolist when facing demand $Q(p)$.

**Remark 3.** *Since $D_t(p) \geq Q(p)$ at any time $t \geq 1$, the monopolist can always obtain the revenue at the monopolist price $REV_{mon} := p_{mon} \cdot q_{mon}$ by choosing price $p_{mon}$. Therefore, we have $REV_t = p_t \cdot q_t \geq REV_{mon}$ for all $t$.*

It turns out that these prices characterize tightly the dynamics for the case of *patient* users ($\delta = 1$), as shown by the next definition and theorem.

**Definition 2** (Eventual Transaction Inclusion, (Minimum) Admission Price). For a given dynamics we consider the following definitions:

- A transaction with price $p$ is *eventually included* if there exists $\Delta_p$ such that, for every $T \geq 1$, there exists some $t$ with $p_t \leq p$ and $T \leq t \leq T + \Delta_p$.

- A price $p$ is called *admission price* if all transactions paying $p$ are eventually included.

- The minimum admission price $p_{map}$ is the smallest admission price such that all transactions paying at least $p_{map}$ are eventually included.

**Theorem 1** (Theorem 1 in [Nis23] restated). *For patient users ($\delta = 1$) and for any strictly decreasing demand function $Q$ and supply $s$ the following holds:*

1. *The dynamics stay always between $p_{ser}$ and $p_{mon}$, that is, prices $p_t$ satisfy $p_{ser} \leq p_t \leq p_{mon}$ for all $t \geq 1$. In particular, transactions paying less than $p_{ser}$ will never be included. At each step $t$, the prices either decrease ($p_t < p_{t-1}$) or they jump up to the monopolist price ($p_t = p_{mon}$).*

2. *Every price larger than $p_{ser}$ is an admission price. Therefore the minimum admission price is exactly $p_{ser}$. Moreover, the dynamics pass through the monopolist price $p_{mon}$ infinitely often.*

*Transactions paying at least $p_{mon}$ are immediately included, and this is tight as there are infinitely steps for which paying less will delay admission to a later step.*

We now go back to the case $\delta \in (0, 1)$. The following example illustrates the dynamics and how different values of $\delta \in (0, 1)$ effect the latter.

**Example 1.** *Let $Q(p) = 1 - p$ for $p \in [0,1]$, $s = 1$ and let $\delta \in (0,1)$. Then, we have the following:*

- *$t = 1$ : The initial demand is $D_1(p) = Q(p)$ and thus we maximize $pD_1(p) = p(1-p)$ which gives $p_1 = 0.5$ as maximizer and $q_1 = D_1(p_1) = 0.5$. This price $p_1$ is the monopolist price and the revenue is $\mathtt{REV}_1 = \mathtt{REV}_{mon} = p_1 q_1 = 0.25$. The pent-up demand is $Z_1(p) = D_1(p) - q_1 = \frac{1}{2} - p$ if $p < p_1$ and zero otherwise.*

- *$t = 2$ : The total demand is $D_2(p) = \delta Z_1(p) + Q(p) = 1 + \delta/2 - (1+\delta)p$ if $p < p_1$ and $Q(p)$ otherwise. We maximize $pD_2(p)$ and get $p_2 = \frac{2+\delta}{4(1+\delta)}$ as maximizer and thus $q_2 = \frac{2+\delta}{4}$. The revenue is $\mathtt{REV}_2 = \frac{(2+\delta)^2}{16(1+\delta)}$. Note that $\mathtt{REV}_2 > \mathtt{REV}_1$ if and only if $\delta > -1$, that is, the revenue from step $t = 2$ is better than the monopolist revenue. The pent-up demand is $Z_2(p) = D_2(p) - q_2 = (2+\delta)/4 - (1+\delta)p$ if $p < p_2$ and zero otherwise.*

- *$t = 3$ : The total demand is $D_3(p) = \delta Z_2(p) + Q(p) = \frac{2\delta + \delta^2 + 4}{4} - (1+\delta+\delta^2)p$ if $p < p_2$ and $Q(p)$ otherwise. Maximization yields $p_3 = \frac{2\delta + \delta^2 + 4}{8(1+\delta+\delta^2)}$. Note that $p_3 < p_2$ if and only if $\delta > 0$. Hence, $q_3 = \frac{2\delta + \delta^2 + 4}{8}$ and the revenue is $\mathtt{REV}_3 = \frac{(2\delta + \delta^2 + 4)^2}{64(1+\delta+\delta^2)}$. Note that $\mathtt{REV}_3 > \mathtt{REV}_1$ if and only if $\delta > \delta^\star := 2\sqrt{2} - 2 \approx 0.828$. That is, if $\delta > \delta^\star$, the price dynamics do not jump and take $p_3$ as above. The pent-up demand is $Z_3(p) = \frac{2\delta + \delta^2 + 4}{8} - (1+\delta+\delta^2)p$ if $p < p_3$ and zero otherwise. However, if $\delta \le \delta^\star$, the revenue will be less than the monopolist revenue. In this case, the price dynamics would jump up to the monopolist price and take $p_3 = p_{mon} = 0.5$ and thus $q_3 = q_{mon} = 0.5$. The pent-up demand would be $Z_3(p) = D_3(p) - q_3 = Q(p) - 0.5 = 0.5 - p$ if $p < p_3$ and zero otherwise.*

*The price dynamics for the first three steps are depicted in Figure 1. Furthermore, Figure 3 shows the price dynamics for 20 steps for different values of $\delta$.*

## 3   Bounds for General Demand Functions

In this section, we analyze the dynamics for different values of $\delta$ and how it affects their behavior. In Section 3.1, we demonstrate that in regimes with a *sufficiently small fraction of expiring transactions* ($\delta < 1$ sufficiently large), the dynamics behaves qualitatively similarly to the case of *no expiring transactions* ($\delta = 1$). The analysis highlights several differences, further investigated in Section 3.2. There we consider finite (possibly small) values of $\delta$ in relation to other parameters, particularly how the "structure" of $Q$ influences the dynamics for $\delta < 1$ compared to the case $\delta = 1$, where the results are uniform for all $Q$ with identical monopolist prices.

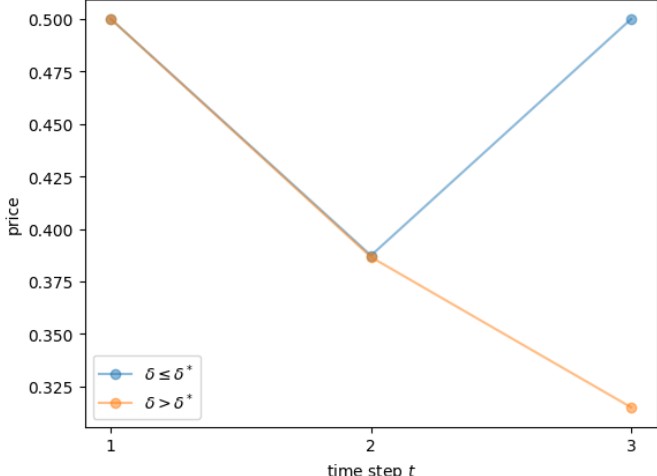

Figure 1: Price dynamics depending on $\delta^\star$ from Example 1. The price at times $t = 2, 3$ are determined by the respective $\delta$. If $\delta$ is below the threshold then the dynamics jump after step $t = 2$. If $\delta$ is above the threshold, the price decreases further for one step.

## 3.1  Upper Bounds on the Admission Price

In this section, we provide a bound on the minimum price which guarantees transactions to be *eventually* included, depending on $\delta$. The main result is summarized by the following definition and the theorem (Definition 3 and Theorem 2).

**Definition 3.** For any continuous decreasing demand function $Q$ and supply $s$ we define the following quantities:

$$\overline{p}_{ser} := \frac{p_{ser} \cdot s}{q_{ser}} \ , \qquad \overline{q}_{ser} := Q(\overline{p}_{ser}) \ , \qquad \overline{\delta}_{ser} := 1 - \frac{q_{ser} - \overline{q}_{ser}}{s} \ . \qquad (8)$$

Moreover, for any $\delta > \overline{\delta}_{ser}$, we let $p_{ser}^{(\delta)}$ be the price such that [6]

$$Q(p_{ser}^{(\delta)}) = q_{ser} - (1 - \delta) \cdot s \ . \qquad (9)$$

**Example 2** (Example 1 continued)**.** *For the setting in Example 1 we observe the minimum admission prices $p_{map}$ for every $\delta \in [0, 1]$ and display it in Figure 2.*

We next state our main result.

**Theorem 2.** *For any strictly decreasing demand function $Q$ and supply $s$ the following holds:*

---

[6]This price exists by continuity and monotonicity of $Q$, and because $Q(\overline{p}_{ser}) = \overline{q}_{ser} = q_{ser} - (1 - \overline{\delta}_{ser}) \cdot s < q_{ser} - (1 - \delta) \cdot s \leq q_{ser} = Q(p_{ser})$, where last inequality is due to $\delta \leq 1$.

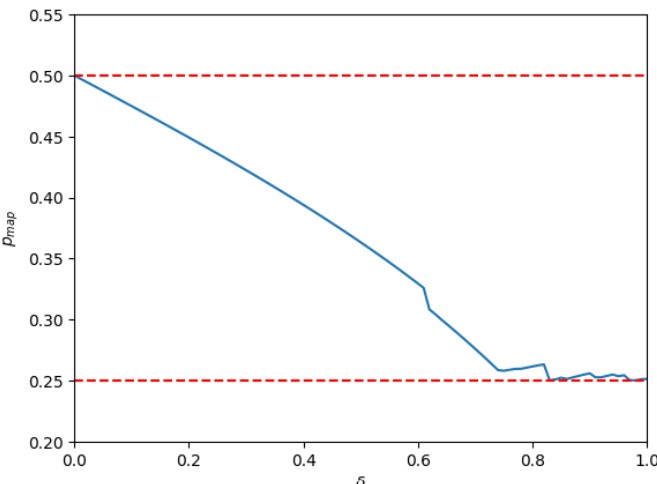

Figure 2: Minimum admission price $p_{map}$ (derived from 100 steps) depending on $\delta$ for daily demand $Q(p) = 1 - p$ and supply $s = 1$.

1. *The minimum admission price is at least $p_{ser}$, and thus transactions paying less than this price will never be included. In particular, the dynamics stay always between $p_{ser}$ and $p_{mon}$, that is, prices $p_t$ satisfy $p_{ser} \leq p_t \leq p_{mon}$ for all $t \geq 1$. Moreover, at each step $t$, the prices either decrease ($p_t < p_{t-1}$) or they jump up to the monopolist price ($p_t = p_{mon}$).*

2. *For every $\delta > \bar{\delta}_{ser}$, the minimum admission price is at most $p_{ser}^{(\delta)}$ defined by (9) which satisfies $p_{ser} < p_{ser}^{(\delta)} < \bar{p}_{ser}$. Moreover, the dynamics pass through the monopolist price $p_{mon}$ infinitely often.*

3. *Every price larger than $p_{ser}$ is an admission price for a sufficiently large $\delta$. That is, for every $p^\star > p_{ser}$, there exists $\delta_{min}(p^\star) < 1$ such that $p^\star$ is an admission price for every $\delta > \delta_{min}(p^\star)$. Moreover, the dynamics pass through the monopolist price $p_{mon}$ infinitely often.*

*Therefore, transactions paying at least $p_{mon}$ are immediately included, and this is tight as there are infinitely steps for which paying less will delay admission to a later step.*

*Proof.* See Appendix A. □

Item 2 in the above theorem provides an upper bound on the minimum admission price, provided $\delta$ being large enough (condition $\delta > \bar{\delta}_{ser}$). This condition on $\delta$ is somehow necessary as implied by the results in the next section, where we prove lower bounds on the minimum admission price, for sufficiently small $\delta$.

## 3.2 Lower Bounds on the Admission Price

In this section, we complement the results in the previous section, by showing that transactions below a certain price will *never* be included, depending on $\delta$.

The monopolist aims to choose a price $p$ maximizing the revenue $\texttt{REV}_t(p) := p \cdot \min(s, D_t(p))$ at the current step $t$. At every time step, there is always the option to choose the monopolist price $p_{mon}$ and receive revenue $\texttt{REV}_{mon}$. We can compare $\texttt{REV}_t(p)$ and $\texttt{REV}_{mon}$ at any $t$ by considering the following function:

$$f_t(p) := p \cdot D_t(p) - p_{mon}q_{mon} \ . \tag{10}$$

Since the revenue for price $p$ satisfies $\texttt{REV}_t(p) \leq p \cdot D_t(p)$, if the function aove is negative for some $p$, it means that the revenue at $p$ is worse than $\texttt{REV}_{mon}$, and therefore the next price $p_t$ cannot be $p$. Observe that evaluating $D_t(p)$ and thus $f_t(p)$ is rather complex because of the "previous history" component involving $q_{t-1}, \ldots, q_1$ – see Equations (4) and (5). We next provide a simpler function to evaluate for a generic $Q$, which still can be used to determine "forbidden" prices for the dynamics:

$$F_t(p) := p \cdot (a_t \cdot Q(p) - (a_t - 1)q_{mon}) - p_{mon}q_{mon} \ , \qquad a_t = \sum_{i=0}^{t} \delta^t \ . \tag{11}$$

**Theorem 3.** *For any $p$ such that $F_t(p) < 0$ it cannot be $p_t = p$.*

*Proof.* We first rewrite Equation (10) using Equation (4) as follows:

$$f_t(p) = p \cdot D_t(p) - p_{mon}q_{mon} = p \cdot (a_t Q(p) - b_t) - p_{mon}q_{mon} \ , \tag{12}$$

where $a_t$ and $b_t$ are defined in (5).

For any $\overline{a}_t \geq a_t$ and any $\underline{b}_t \leq b_t$ we can obtain an upper bound on $f_t(\cdot)$:

$$f_t(p) \leq \overline{f}_t(p) := p \cdot (\overline{a}_t Q(p) - \underline{b}_t) - p_{mon}q_{mon} \ . \tag{13}$$

For any $p$ such that $\overline{f}_t(p) < 0$, we obviously have $f_t(p) < 0$, which implies that it cannot be $p_t = p$. Indeed, definition of $f_t(p) < 0$ implies $p \cdot D_t(p) < p_{mon}q_{mon}$ and hence $p$ will not be taken.

Next observe that, since $p_t \leq p_{mon}$, the monotonicity of $Q$ implies $q_t \geq q_{mon}$, thus implying

$$b_t \geq (a_t - 1)q_{mon} =: \underline{b}_t \ . \tag{14}$$

Finally, observe that for $\underline{b}_t$ we have $\overline{f}_t(p)$ is exactly $F_t$ in (11). This completes the proof. $\qquad \square$

### 3.2.1 A first example

We next apply the result in Theorem 3 to one of the simplest demand functions and show that, even in this case, price $p_{ser}$ is not a tight bound for the minimum admission price.

**Proposition 1** (Lower bound)**.** *For demand function $Q(p) = 1 - p$, the price $p^\star := \frac{1-\delta}{2}$ is a lower bound for the price dynamics.*

*Proof.* Theorem 3 yields a lower bound for the smallest $p$ of the dynamics. Since $p_{mon} = \frac{1}{2} = q_{mon}$ we have

$$F_t(p) = \left( \frac{(1 - \delta^t)(1 - p)}{1 - \delta} - \frac{\frac{1-\delta^t}{1-\delta} - 1}{2} \right) p - \frac{1}{4} \tag{15}$$

which has two roots. One root is at $\frac{1}{2}(= p_{mon})$ and the other one is

$$p_t^\star = \frac{1 - \delta}{2(1 - \delta^t)} \ . \tag{16}$$

We can make the following observations:

1. For $t \to \infty$ and fixed $\delta \in (0, 1)$ we have $p_t^\star \to p^\star := \frac{1-\delta}{2}$ meaning that the dynamics cannot go below this minimum price $p^\star$.

2. For $\delta \to 1$ and for fixed $t \geq 2$ we have $p_t^\star \to \frac{1}{2t}$ meaning that the dynamics cannot be below $\frac{1}{2t}$ at step $t$.

This completes the proof. $\qquad\square$

**Remark 4.** *According to the previous result, for $\delta = \frac{1}{2}$ the dynamics never go below $\frac{1}{4}$. That is, the minimum admission price is at least $\frac{1}{4}$ and transactions paying less than this price are never admitted. We note that this bound is not tight, as the dynamics for $\delta = 0.5$ never passes value $\approx 0.363$ (see Figure 3). Figure 3 further shows the dynamics for different values of $\delta$.*

### 3.2.2 The Admission Price Must Depend on $Q$

In this section, we consider the following class $\mathcal{Q}_\epsilon$ of daily demand functions:

$$Q(p) = \begin{cases} \frac{1}{2} + \epsilon - 2\epsilon p, & 0 \leq p \leq \frac{1}{2} \\ 1 - p, & \frac{1}{2} \leq p \leq 1 \end{cases} \tag{17}$$

A function $Q$ of this class $\mathcal{Q}_\epsilon$ is depicted in Figure 4. In particular, note that the slope of the function on the interval $[0, 0.5]$ is depending on $\epsilon \geq 0$. On the remaining interval $[0.5, 1]$ the function is just $1 - p$.

A few observations are in place.

**Obs 1.** Theorem 2 provides an upper bound on the minimum admission price if $\delta > \bar{\delta}_{ser}$. According to (8), this condition is equivalent to $\frac{q_{ser} - \bar{q}_{ser}}{1 - \delta} > s$.[7]

---

[7]For $\epsilon = \frac{1}{2}$ and $s = 1$, the condition in **Obs 1** boils down to $\delta > 11/12 \approx 0.917$,

$$\frac{Q(p) - Q(p')}{1 - \delta(\epsilon)} > s \iff \delta > 11/12 \approx 0.917.$$

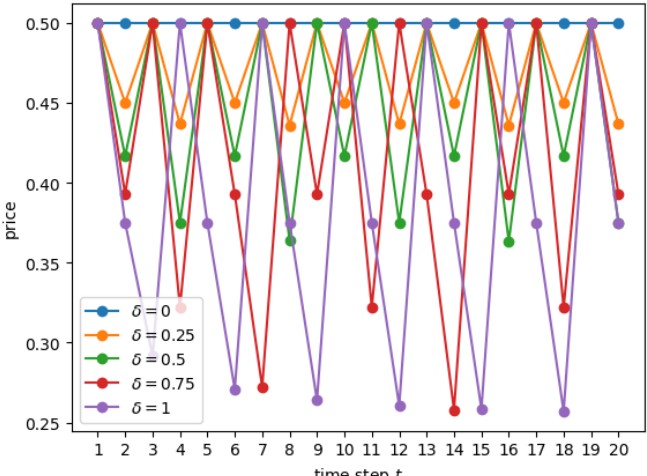

Figure 3: The dynamics for $Q(p) = 1 - p$ and $\delta \in [0, 0.25, 0.5, 0.75, 1]$ when supply $s = 1$. Note that there is an additional step before the jumps for $\delta = 1$. This is in fact true for any $\delta > \delta^\star = 2\sqrt{2} - 2 \approx 0.828$ (see Example 1). The smallest prices for each $\delta$ correspond to the respective minimum admission prices. (Note that this figure displays only 20 iterations and for a price dynamic to reach its minimum it may take more steps.)

**Obs 2.** For the type of demand function $Q \in \mathcal{Q}_\epsilon$, Theorem 2 may not apply, unless **Obs 1** is satisfied. This also depends on the value of $\epsilon$.

**Obs 3.** The lower bound $p_{ser}$ is not tight (Remark 4 deals with $\epsilon = \frac{1}{2}$).

These observations naturally suggest to obtain *lower bounds* on the minimum admission price for the class of functions above.

**Example 3.** *Theorem 2 applies only for $\delta$ that are large enough ($\delta > \bar{\delta}_{ser}$). The necessary lower bound for $\delta$ for demand function of class $\mathcal{Q}_\epsilon$ is calculated below , that is, for the demand functions as in Equation (17) and $s = 1$, we have (by Definition 3)*

$$p_{mon} = \frac{1}{2} = q_{mon} \; , \qquad p_{ser} = \frac{1}{4} \; , \qquad q_{ser} = \frac{1 + \epsilon}{2} \; , \quad (18)$$

$$\bar{p}_{ser} = \frac{1}{2(1 + \epsilon)} \; , \quad \bar{q}_{ser} = \frac{1}{2} + \frac{\epsilon^2}{1 + \epsilon} \; , \quad \delta_{\min}(\bar{p}_{ser}) = 1 - \frac{\epsilon}{2} + \frac{\epsilon^2}{1 + \epsilon} \; , \quad (19)$$

$$p_{ser}^{(\delta)} = \frac{1}{4} + \frac{1 - \delta}{\epsilon} \; , \tag{20}$$

*thus implying that for $Q(p) = 1 - p$, and $\epsilon = \frac{1}{2}$, we have*

$$p_{ser}^{(\delta)} = \frac{1}{4} + 2(1 - \delta) \qquad \text{for all } \delta > \bar{\delta}_{ser} = \delta_{\min}(\bar{p}_{ser}) = 11/12 \; . \tag{21}$$

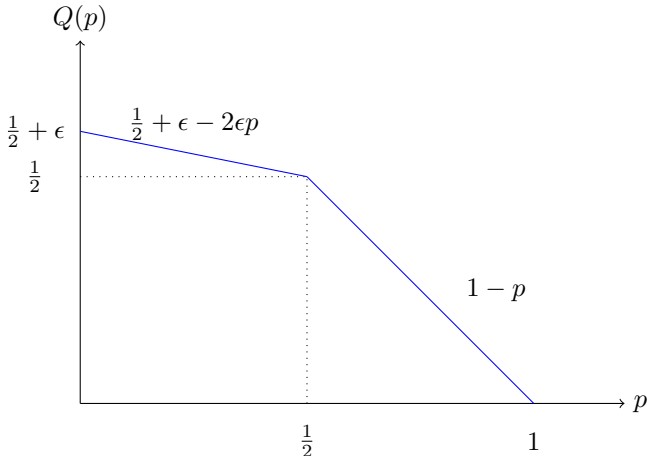

Figure 4: Daily demand function of the class $\mathcal{Q}_\epsilon$.

Next, we state three results for the class of demand functions $\mathcal{Q}_\epsilon$. First, for any $\delta$ we can find a strictly decreasing demand function such that the price dynamics are stuck at $p_{mon}$. Second, for any strictly decreasing demand function we find a $\delta$ such that the same holds. Finally, we find a lower bound for the price dynamics if a certain condition on the demand function are met.

**Theorem 4.** *The following holds:*

1. *For every $\delta$ there exists a strictly decreasing $Q \in \mathcal{Q}_\epsilon$ such that the price dynamics stays at $p_{mon}$, i.e.*

$$p_t = p_{mon}, \quad \text{for all } t. \tag{22}$$

2. *Conversely, for any strictly decreasing $Q \in \mathcal{Q}_\epsilon$ we find a $\delta$ such that Equation (22) holds.*

3. *The price $p^\star = \frac{1-\delta}{4\epsilon} < p_{mon}$ is a new lower bound for the price dynamics if $\epsilon \geq \frac{1-\delta}{2(1-\delta^t)}$ for all $t$.*

*Proof.* We apply Theorem 3 to $Q \in \mathcal{Q}_\epsilon$ and consider the corresponding function

$$F_t(p) = p \cdot \left( \frac{(1-\delta^t)\left(-2\epsilon p + \epsilon + \frac{1}{2}\right)}{1-\delta} - \frac{\frac{1-\delta^t}{1-\delta} - 1}{2} \right) - \frac{1}{4}, \tag{23}$$

which has two roots. One root is at 0.5 and the second root is

$$p_t^\star := \frac{\delta - 1}{4\delta^t \epsilon - 4\epsilon}. \tag{24}$$

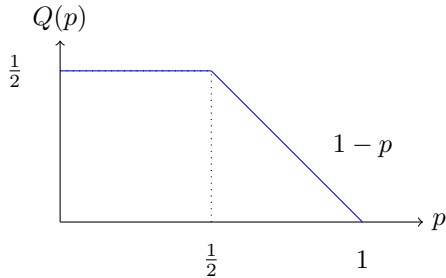

Figure 5: Daily demand function of class $\mathcal{Q}_\epsilon$ for $\epsilon = 0$.

Note that

$$p_t^\star \overset{t \to \infty}{\to} \frac{1-\delta}{4\epsilon} =: p^\star. \tag{25}$$

Furthermore, we have that $p_t^\star < p^\star$. Therefore, for $t \to \infty$ and $\epsilon < \frac{1-\delta}{2}$ this root $p_t^\star$ is bigger than the monopolist price and this implies that the dynamics never goes below $p_{mon}$. Similarly, for a fixed $\epsilon$ (i.e. a given $Q \in \mathcal{Q}_\epsilon$) we can find the $\delta$ that satisfies the latter condition, the dynamics never go below $p_{mon}$.

Furthermore, if $\epsilon \geq \frac{1-\delta}{2(1-\delta^t)}$ for all $t$, then the root in (24) is below 0.5, which gives us a lower bound for the price dynamics. Alternatively, note that, if $\epsilon \geq \frac{1-\delta}{2(1-\delta^1)} = \frac{1}{2}$ holds, then $\epsilon \geq \frac{1-\delta}{2(1-\delta^t)}$ for all $t$. In particular, for a fixed $\epsilon$ we can find two intervals for $\delta$ such that we end up with a lower bound less than $p_{mon}$ on one interval and a lower bound equal to $p_{mon}$ on the other interval. $\quad\square$

Next, we discuss the case when the demand function is not strictly decreasing, but rather constant for some interval of prices.

**Remark 5.** *For $\epsilon = 0$, we have a piecewise constant demand function, see Figure 5. For any $s \geq 0.5$, the price dynamics stays at $p_{mon} = 0.5$ which can be verified by observing that for any $p < p_{mon}$ and any $t \geq 1$ we have $D_t(p) = \frac{1}{2}$. Therefore, to maximize revenue the monopolist will always choose $p_t = p_{mon}$. Note that in Nisan's setting [Nis23], with $\delta = 1$, the same happens, as this is independent of $\delta$. All demand is supplied in each round. In particular, the lower bound has to be adjusted to be the largest $p$ satisfying such that $Q(p_{ser}) = Q(p_{mon}q_{mon}/s)$. In this example, $p_{mon}q_{mon} = \frac{1}{4}$ and for $s \geq 0.5$, $Q(p_{mon}q_{mon}/s) = 0.5$. Clearly, the largest $p$ s.t. $Q(p) = 0.5$ is $p = 0.5(=: p_{ser})$ which is the new lower bound (and equal to the upper bound $p_{mon} = 0.5$).*

### 3.2.3 General demand function $Q$

Let $Q$ be a general demand function. Furthermore, let $Q$ be upper bounded by a constant $K$, i.e. $Q(p) \leq K$ for all $p$. We provide a lower bound that is higher than $p_{ser}$.

**Proposition 2.** *Let $Q$ be a general demand function such that $Q(p) \leq K$ for some constant $K$. Furthermore, let $K$ satisfy $K \leq s + \delta(q_{mon} - s)$. Then, the price*

$$p^\star := \frac{p_{mon}q_{mon}}{\frac{K}{1-\delta} + q_{mon} - \frac{q_{mon}}{1-\delta}}, \tag{26}$$

*is a lower bound for the price dynamics. In particular, $p^\star \geq p_{ser}$.*

*Proof.* Similar to earlier calculations, we define

$$\overline{f}_t(p) := p(a_t Q(p) - (a_t - 1)q_{mon}) - p_{mon}q_{mon}, \tag{27}$$

and

$$\overline{\overline{f}}_t(p) := p(a_t K - (a_t - 1)q_{mon}) - p_{mon}q_{mon}, \tag{28}$$

where $\overline{\overline{f}}_t(p)$ uses the upper bound on the demand function. Note that $f_t(p) \leq \overline{f}_t(p) \leq \overline{\overline{f}}_t(p)$ where $f_t(p)$ is as in Equation (10). Being linear in $p$, note that $\overline{\overline{f}}_t(p)$ has a root at

$$\overline{\overline{p}}_t = \frac{p_{mon}q_{mon}}{a_t K - (a_t - 1)q_{mon}} \xrightarrow{t \to \infty} \frac{p_{mon}q_{mon}}{\frac{K}{1-\delta} + q_{mon} - \frac{q_{mon}}{1-\delta}} =: \overline{\overline{p}}. \tag{29}$$

Hence this $\overline{\overline{p}}$ is a new lower bound if $K \leq s + \delta(q_{mon} - s)$ since then $\overline{\overline{p}} \geq p_{ser}$. The proof is completed as $\overline{\overline{p}}$ corresponds to $p^\star$ in the theorem statement. $\qquad\square$

One example where the conditions in Proposition 2 are satisfied follows.

**Example 4.** *For $s = \frac{Q(0)}{1-\delta}$ and $K = Q(0)$, the condition above ($K \leq s + \delta(q_{mon} - s)$) is satisfied and $\overline{\overline{p}} = \frac{p_{mon}q_{mon}}{s - \frac{\delta}{1-\delta}q_{mon}}$ which is larger than $p_{ser}$.*

# 4 Conclusion and Future Work

In this paper we analyzed the fluctuations in prices under the monopolistic pricing mechanism with quasi-patient users. In particular, only a fraction of unsupplied transactions remains in the mempool after each block. A transaction paying at least the minimum admission price will be eventually included in a block. We provided upper and lower bounds for the admission price. Furthermore, we compared our bounds to the bounds achieved in [Nis23] and highlight the differences.

The analysis of strategic agents is left to future research. Our analysis concentrated around the question of a monopolist maximizing myopically considering the current round only. An interesting future direction is to consider agents maximizing over multiple rounds given that agents expect to be the block leader for several consecutive blocks. Furthermore it remains to show properties of the minimum admission price $p_{map}$ depending on $\delta$. In fact, for the particular linear demand function ($Q(p) = 1 - p$), in Figure 2 we observe a decreasing $p_{map}$ up until approx. 0.8 and some kind of up and down after 0.8. A reason for this behavior can be a lack of steps, that is, the price dynamics for some $\delta$ might need more steps to attain its minimum prices.

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

# A Postponed Proofs

This section provides lemmas (and its proofs) that will be used to prove Theorem 2. Note that the lemmas follow the style of [Nis23] but are adapted to our setting.

Throughout this section we use the quantity $a_t$ defined in Equation (5), which we rewrite here for convenience:

$$a_t = 1 + \delta + \cdots + \delta^{t-1} = \frac{1 - \delta^t}{1 - \delta} , \qquad\qquad \delta \in [0, 1) . \qquad (30)$$

**Lemma 1.** *For $p < p'$ and any $t$:*

$$D_t(p) - D_t(p') \leq a_t \cdot (Q(p) - Q(p'))$$

*where $a_t$ is defined as in* (30).

*Proof.* Fix any $t \geq 1$. We first show that the following holds:

$$Z_{t-1}(p) - Z_{t-1}(p') \leq D_{t-1}(p) - D_{t-1}(p') . \qquad (31)$$

We distinguish the following cases:

1. $(p_{t-1} < p < p'.)$ We have that $Z_{t-1}(p) = Z_{t-1}(p') = 0$ and hence $D_t(p) - D_t(p') \leq [Q(p) - Q(p')]$

2. $(p < p' < p_{t-1}.)$ We have that $Z_{t-1}(p) = D_{t-1}(p) - q_{t-1}$ and $Z_{t-1}(p') = D_{t-1}(p') - q_{t-1}$. Together, $Z_{t-1}(p) - Z_{t-1}(p') = D_{t-1}(p) - D_{t-1}(p')$ and hence $D_t(p) - D_t(p') = \delta(D_{t-1}(p) - D_{t-1}(p')) + [Q(p) - Q(p')]$. Iteratively, we get $D_t(p) - D_t(p') = (1 + \delta + \delta^2 + \cdots + \delta^{t-1})[Q(p) - Q(p')]$.

3. $(p < p_{t-1} < p'.)$ We have that $Z_{t-1}(p) = D_{t-1}(p) - q_{t-1}$ and $Z_{t-1}(p') = 0$. Also note that $D_{t-1}(p') \leq q_{t-1} = D_{t-1}(p_{t-1})$. Hence, $Z_{t-1}(p) - Z_{t-1}(p') = D_{t-1}(p) - q_{t-1} \leq D_{t-1}(p) - D_{t-1}(p')$. And therefore we have that,

$$D_t(p) - D_t(p') \leq \delta(D_{t-1}(p) - D_{t-1}(p')) + [Q(p) - Q(p')] \qquad (32)$$

   Iteratively, we get again $D_t(p) - D_t(p') \leq (1 + \delta + \delta^2 + \cdots + \delta^{t-1})[Q(p) - Q(p')]$

We next prove the lemma by induction on $t$.

- *Base case* $(t = 1)$: $D_1(p) - D_1(p') = Q(p) - Q(p') \leq 1[Q(p) - Q(p')]$

- *Inductive step*: Assume the claim is true for $t - 1$. Then we have

$$
\begin{aligned}
D_t(p) - D_t(p') &= \delta(Z_{t-1}(p) - Z_{t-1}(p')) + [Q(p) - Q(p')] \\
&\leq \delta(D_{t-1}(p) - D_{t-1}(p')) + [Q(p) - Q(p')] \\
&\leq \delta((1 + \delta + \delta^2 + \cdots + \delta^{t-2})[Q(p) - Q(p')]) + [Q(p) - Q(p')] \\
&= (1 + \delta + \delta^2 + \cdots + \delta^{t-1})[Q(p) - Q(p')]
\end{aligned}
$$

where we used Equation (31) in the second line.

$\square$

**Lemma 2.** *For $p < p'$ and for all $T$ and $t > T$:*

$$D_t(p) - D_t(p') \leq a_{t-T} \cdot (Q(p) - Q(p')) + \delta^{t-T} \cdot (Z_T(p) - Z_T(p')) , \qquad (33)$$

*where $a_t$ is defined as in (30).*

*Proof.* Similar to the proof above. By repeating Equation (32), $(t-T-1)$-times and in another step we just replace $D_T(p) - D_T(p')$ with its definition, we end up with the expression:

$$D_t(p) - D_t(p') \leq \delta^{t-T}(Z_T(p) - Z_T(p')) + (1 + \delta + \cdots + \delta^{t-T-1})[Q(p) - Q(p')]. \quad (34)$$

By induction, it holds for all $t$:

- $t = 1$: $D_1(p) - D_1(p') = \delta(Z_0(p) - Z_0(p')) + [Q(p) - Q(p')] = [Q(p) - Q(p')]$

- Assume the claim is true for $t - 1$

- $t - 1 \to t$:

$D_t(p) - D_t(p')$
$= \delta(Z_{t-1}(p) - Z_{t-1}(p')) + [Q(p) - Q(p')]$
$\leq \delta(D_{t-1}(p) - D_{t-1}(p')) + [Q(p) - Q(p')]$
$\leq \delta(\delta^{t-T-1}(Z_T(p) - Z_T(p')) + (1 + \delta + \cdots + \delta^{t-T-2})[Q(p) - Q(p')]) + [Q(p) - Q(p')]$
$= \delta^{t-T}(Z_T(p) - Z_T(p')) + (1 + \delta + \cdots + \delta^{t-T-1})[Q(p) - Q(p')].$

$\square$

**Lemma 3.** *For all $T$ and $t > T$, if for all $t'$ such that $T < t' < t$ we also have that $p_{t'} \geq p' > p$ then in fact we have equality*

$$D_t(p) - D_t(p') = a_{t-T} \cdot (Q(p) - Q(p')) + \delta^{t-T}(Z_T(p) - Z_T(p')) \qquad (35)$$

*where $a_t$ is defined as in (30).*

*Proof.* For $p_{t-1} \geq p' > p$ we have that $Z_{t-1}(p) = D_{t-1}(p) - q_{t-1}$ and $Z_{t-1}(p') = D_{t-1}(p') - q_{t-1}$. Therefore, $D_t(p) - D_t(p') = \delta(Z_{t-1}(p) - Z_{t-1}(p')) + [Q(p) - Q(p')] = \delta(D_{t-1}(p) - D_{t-1}(p')) + [Q(p) - Q(p')]$. By induction the claim holds.

- $t = 1$: $D_1(p) - D_1(p') = \delta(Z_0(p) - Z_0(p')) + [Q(p) - Q(p')] = [Q(p) - Q(p')]$

- Assume the claim is true for $t - 1$

- $t - 1 \to t$:

$D_t(p) - D_t(p')$
$= \delta(Z_{t-1}(p) - Z_{t-1}(p')) + [Q(p) - Q(p')]$
$= \delta(D_{t-1}(p) - D_{t-1}(p')) + [Q(p) - Q(p')]$
$= \delta((1 + \delta + \cdots + \delta^{t-T-2})[Q(p) - Q(p')] + \delta^{t-T-1}(Z_T(p) - Z_T(p'))) + [Q(p) - Q(p')]$
$= (1 + \delta + \cdots + \delta^{t-T-1})[Q(p) - Q(p')] + \delta^{t-T}(Z_T(p) - Z_T(p')).$

$\square$

**Lemma 4.** *For all $T$ such that $p_T \le p < p'$ (or $T = 0$) and all $t > T$, we have*

$$D_t(p) - D_t(p') \le a_{t-T} \cdot (Q(p) - Q(p')) \tag{36}$$

*where $a_t$ is defined as in (30). Furthermore, for all $T$ such that $p_T \le p < p'$ (or $T = 0$), if for all $t'$ such that $T < t' < t$ we also have that $p_{t'} \ge p' > p$, then the equation above holds with equality, i.e.*

$$D_t(p) - D_t(p') = a_{t-T} \cdot (Q(p) - Q(p')) . \tag{37}$$

*Proof.* Apply the previous two lemmas and note that $Z_T(p) = Z_T(p') = 0$ since $p_T \le p < p'$. $\square$

The proofs of Lemmas 5 and 6 below are the same as in [Nis23], and we restate them here for the sake of completeness.

**Lemma 5.** *For every $t$ it holds that $p_t \ge p_{ser}$.*

*Proof.* The maximum revenue that is achievable from a price $p$ is $p \cdot s$. For $p < p_{ser}$ we have that $p \cdot s < p_{ser} \cdot s = p_{mon} \cdot q_{mon}$, and the latter revenue can be achieved at any step using the monopolist price. $\square$

**Lemma 6.** *For every $t$ either $p_t = p_{mon}$ or $p_t < p_{t-1}$.*

*Proof.* For $p \ge p_{t-1}$ we have that $D_t(p) = Q(p)$ so the maximal revenue obtained by possible $p \ge p_{t-1}$ is exactly the monopolist's revenue that is obtained at $p_t = p_{mon}$ (we assume that ties in maximum revenue are broken consistently). So, unless $p_t = p_{mon}$, we must obtain the maximum revenue for some $p < p_{t-1}$. $\square$

The next lemma provides a sufficient condition for the price to decrease.

**Lemma 7.** *Assume that for some $p > p_{ser}$ we have that $D_t(p) \ge s$, then*

(i) $p_t < p_{t-1}$, *and*

(ii) $Q(p_t) - Q(p_{t-1}) \ge (a_{t-1})^{-1} s \frac{(p-p_{ser})}{p_{mon}}$

*where $a_t$ is defined as in (30). Furthermore, if for some $T < t$ we had $p_T \le p_t$ then, $Q(p_t) - Q(p_{t-1}) \ge (a_{t-1-T})^{-1} s \frac{(p-p_{ser})}{p_{mon}}$.*

*Proof.* We observe the following:

- We cannot have $p_t = p_{mon}$, as the revenue obtained from $p$ would be higher: $ps > p_{ser}s = p_{mon}q_{mon}$.

- As $p_t$ gives better revenue than $p$, i.e. we have that $p_t q_t = p_t D_t(p_t) \ge ps = (p - p_{ser})s + p_{ser}s = (p - p_{ser})s + p_{mon}q_{mon}$.

- Separating the total demand at time $t$ to its two components we get

$$p_t D_t(p_t) = \delta p_t Z_{t-1}(p_t) + p_t Q(p_t)$$
$$\leq {}^8 \delta p_t Z_{t-1}(p_t) + p_{mon} q_{mon}$$
$$\leq \delta p_{mon} Z_{t-1}(p_t) + p_{mon} q_{mon}.$$

- Putting these together we get that

$$(p - p_{ser})s + p_{mon} q_{mon} \leq p_t D_t(p_t) \leq \delta p_{mon} Z_{t-1}(p_t) + p_{mon} q_{mon}, \quad (38)$$

that is

$$(p - p_{ser})s \leq \delta p_{mon} Z_{t-1}(p_t) . \quad (39)$$

- Now,

$$Z_{t-1}(p_t) = Z_{t-1}(p_t) - Z_{t-1}(p_{t-1})$$
$$\leq D_{t-1}(p_t) - D_{t-1}(p_{t-1})$$
$$\leq (1 + \delta + \ldots + \delta^{t-2})[Q(p_t) - Q(p_{t-1})],$$

where we used Lemma 1 in the last step.

- Hence, it follows that

$$(p - p_{ser})s \leq p_{mon}(1 + \delta + \ldots + \delta^{t-2})[Q(p_t) - Q(p_{t-1})] \quad (40)$$

and therefore,

$$[Q(p_t) - Q(p_{t-1})] \geq (1 + \delta + \ldots + \delta^{t-2})^{-1} s \frac{(p - p_{ser})}{p_{mon}}. \quad (41)$$

- The second part of the lemma is similar.

$\square$

The next lemmas assume some extra condition (which is necessary in our setting), that is, we consider pair of prices such that

$$\frac{1}{1 - \delta} \cdot (Q(p) - Q(p')) > s \qquad\qquad p < p' . \qquad\text{(gap)}$$

**Lemma 8.** *For every $p < p'$ satisfying* (gap) *there exists $\Delta_0$ s.t. for all $T$ and all $\Delta \geq \Delta_0$ we have that either (a) there exists $T \leq t \leq T + \Delta$ with $p_t < p'$ or (b) $D_{T+\Delta}(p) \geq s$.*

---

[8] In this step we use that $p_{mon}$ maximizes $pQ(p)$ and $q_{mon} = Q(p_{mon})$. Also we know that $p_t < p_{mon}$

*Proof.* Let us observe that, if $p_t \geq p'$ for all $T \leq t \leq T + \Delta$, then using Lemma 3 we get

$$
\begin{aligned}
D_{T+\Delta}(p) &\geq D_T(p) - D_T(p') \\
&\stackrel{Lem\ 3}{=} a_\Delta \cdot (Q(p) - Q(p')) + \delta^\Delta (Z_T(p) - Z_T(p')) \\
&\geq a_\Delta \cdot (Q(p) - Q(p')) = (1 + \delta + \cdots + \delta^{\Delta-1}) \cdot (Q(p) - Q(p')) .
\end{aligned}
$$

We next show that, for sufficiently large $\Delta$, the latter quantity must exceed $s$. That is, we can find $\Delta_0$ such that $(1 + \delta + \cdots + \delta^{\Delta_0 - 1}) \cdot (Q(p) - Q(p')) = s$:

$$
\begin{aligned}
\frac{1 - \delta^{\Delta_0}}{1 - \delta} = \frac{s}{Q(p) - Q(p')} \quad &\Leftrightarrow \quad \delta^{\Delta_0} = 1 - \frac{s(1-\delta)}{Q(p) - Q(p')} \\
&\Leftrightarrow \quad \Delta_0 = \ln\left((1-\delta) - \frac{s(1-\delta)}{Q(p) - Q(p')}\right) \\
&\Leftrightarrow \quad \Delta_0 = \ln(1-\delta) + \ln\left(1 - \frac{s}{Q(p) - Q(p')}\right) .
\end{aligned}
$$

This completes the proof. $\qquad\square$

**Lemma 9.** *For every $p^\star > p_{ser}$ such that*

$$
\delta > \delta_{\min}(p^\star) := 1 - \frac{Q(p_{ser}) - Q(p^\star)}{s} \tag{42}
$$

*the following holds. There exists $\Delta$ such that for every $T$ there exists some $T < t \leq T + \Delta$ with $p_t \leq p^\star$.*

*Proof.* By contradiction, assume $T = 0$ or $T$ such that $p_t > p^\star$ for all $t \in [T, T+\Delta]$, for all $\Delta$. Let

$$
p := \frac{p^\star + p_{ser}}{2} \tag{43}
$$

so $p_{ser} < p < p^\star$ and $p^\star - p_{ser} = 2(p - p_{ser})$. Next observe that (42) says that the condition (gap) required in Lemma 8 holds for $p = p_{ser}$ and $p' = p^\star$. Hence, there exists $\Delta_0$ after which $D_t(p) \geq s$ for all $t \geq T + \Delta_0$ until the first time that $p_t \leq p^\star$. Fix $\Delta > \Delta_0$ [9] such that $D_t(p) \geq s$ for all $t \in [T + \Delta_0, T + \Delta]$. By Lemma 7 we get a sequence of decreasing prices

$$
p_{T+\Delta_0} > p_{T+\Delta_0+1} > \cdots > p_{T+\Delta} \tag{44}
$$

such that, for all $t \in [T + \Delta_0 + 1, T + \Delta]$, we have

$$
Q(p_t) - Q(p_{t-1}) \geq \frac{1}{a_{t-1}} \cdot \frac{p - p_{ser}}{p_{mon}} \cdot s . \tag{45}
$$

---

[9]It may happen that such a $\Delta$ does not exist. In particular, if $D_t(p) \geq s$ only for $t = T + \Delta_0$ and for $t = T + \Delta_0 + 1$ we may have $p_t \leq p^\star$.

Hence, using $a_t = \frac{1-\delta^t}{1-\delta}$, we have

$$
\begin{aligned}
Q(p_{T+\Delta}) - Q(p_{T+\Delta_0}) &\geq \sum_{t=T+\Delta_0+1}^{T+\Delta} \frac{1}{a_{t-1}} \cdot \frac{p - p_{ser}}{p_{mon}} \cdot s \\
&= \left( \frac{1-\delta}{1-\delta^{\Delta_0}} + \cdots + \frac{1-\delta}{1-\delta^{\Delta-1}} \right) \cdot \frac{p - p_{ser}}{p_{mon}} \cdot s \\
&\geq \frac{1-\delta}{1-\delta^{\Delta_0}} \cdot (\Delta - \Delta_0) \cdot \frac{p - p_{ser}}{p_{mon}} \cdot s .
\end{aligned}
$$

Next observe that $Q(p_{T+\Delta}) - Q(p_{T+\Delta_0}) \leq Q(p_{ser}) - Q(p_{mon})$, since all prices are between $p_{ser}$ and $p_{mon}$, and $Q$ is decreasing. We therefore get a contradiction if this inequality holds:

$$
Q(p_{ser}) - Q(p_{mon}) < \frac{1-\delta}{1-\delta^{\Delta_0}} \cdot (\Delta - \Delta_0) \cdot \frac{p - p_{ser}}{p_{mon}} \cdot s .
$$

This condition is true whenever

$$
\Delta > \Delta_0 + Q(p_{ser}) - Q(p_{mon}) \frac{p_{mon}(1 - \delta^{\Delta_0})}{s(p - p_{ser})(1 - \delta)}
$$

and thus we have the contradiction for such $\Delta$. We conclude that we must have $p_t \leq p^\star$. $\qquad\square$

We next provide a technical lemma relating $\delta$ and values of $p^\star$ for which the dynamics must take the monopolist price infinitely often.

**Lemma 10.** *Suppose there exist two values $p^\star > p_{ser}$ and $q^\star > q_{ser}$ satisfying (42) and the following inequality:*[10]

$$
p^\star q^\star < p_{mon} q_{mon} = p_{ser} \cdot s . \tag{46}
$$

*Then, there exist infinitely many $t$ such that $p_t = p_{mon}$.*

*Proof.* By contradiction, assume there is a last time $\ell \geq 1$ such that the dynamics takes the monopolist price. We then observe the following:

1. We have a sequence of decreasing prices (Lemma 6)

$$
p_\ell > p_{\ell+1} > \cdots \tag{47}
$$

   satisfying $p_t \geq p_{ser}$ for all $t \geq \ell$ (Lemma 5).

2. By Lemma 9 there is some $T_0 \geq \ell$ such that, for all $t \geq T_0$, we have

$$
p_t < p^\star , \qquad p_t q_t \geq p_{mon} q_{mon} = p_{ser} \cdot s \quad \overset{(46)}{\Rightarrow} \quad q_t > q^\star \tag{48}
$$

   where the second inequality holds because the dynamics prefers $p_t$ to $p_{mon}$.

---

[10]Note that $q^\star \neq Q(p^\star)$ and in particular $q^\star > q_{ser} = Q(p_{ser}) > Q(p^\star)$.

3. The monotonicity of $Q$ and $p_t \geq p_{ser}$ imply $q_t \leq q_{ser} < q^\star$, thus contradicting (48).

This completes the proof. □

We next turn our attention to the existence of $p^\star$ and $q^\star$.

**Lemma 11.** *For every continuous $Q$ such that the equilibrium revenue is less than the monopolist revenue, the following hold:*

1. *Quantities in (8) satisfy $\bar{p}_{ser} > p_{ser}$ and $\bar{q}_{ser} < q_{ser}$.*

2. *For every $p^\star > p_{ser}$ satisfying $p^\star < \bar{p}_{ser}$ condition (46) in Lemma 10 holds for some $q^\star > q_{ser}$.*

3. *For every $\delta > \bar{\delta}_{ser}$, there exists $p^\star > p_{ser}$ satisfying $p^\star < \bar{p}_{ser}$ such that condition (42) in Lemma 9 holds. In particular, this holds true for any $p_{ser}^{(\delta)} < p^\star < \bar{p}_{ser}$.*

*Proof.* We distinguish the three parts:

1. As shown in [Nis23], if the equilibrium revenue is smaller than the monopolist revenue, then $q_{ser} < s$. Indeed, the equilibrium is given by the price $p_{eq}$ such that $q_{eq} := Q(p_{eq}) = s$, and $Rev_{eq} = p_{eq}s < Rev_{mon} = p_{mon}q_{mon} = p_{ser}s$, thus implying $p_{eq} < p_{ser}$. Hence,

2. Since $p^\star q_{ser} < p_{ser}s$, we have that $p^\star q_{ser} \cdot \rho < p_{ser}s$ for sufficiently small $\rho > 1$. Hence, $q^\star = q_{ser}\rho > q^\star$ satisfy the desired conditions.

3. We first show that $p_{ser}^{(\delta)} < \bar{p}_{ser}$. Observe that

$$\delta_{\min}(p_{ser}^{(\delta)}) = 1 - \frac{q_{ser} - Q(p_{ser}^{(\delta)})}{s} \tag{49}$$

$$\overset{(9)}{=} 1 - \frac{q_{ser} - (q_{ser} - (1 - \delta) \cdot s)}{s} = \delta \tag{50}$$

$$> \bar{\delta}_{ser} = 1 - \frac{q_{ser} - Q(\bar{p}_{ser})}{s} . \tag{51}$$

Therefore, $Q(p_{ser}^{(\delta)}) > Q(\bar{p}_{ser})$ and, by monotonicity of $Q$, we have $p_{ser}^{(\delta)} < \bar{p}_{ser}$. For any $p^\star$ such that $p_{ser}^{(\delta)} < p^\star < \bar{p}_{ser}$ we have $Q(p^\star) < Q(p_{ser}^{(\delta)})$ and therefore

$$\delta_{\min}(p^\star) = 1 - \frac{q_{ser} - Q(p^\star)}{s} < 1 - \frac{q_{ser} - Q(p_{ser}^{(\delta)})}{s} \overset{(50)}{=} \delta . \tag{52}$$

This completes the proof. □

We restate Theorem 2.

**Theorem 2.** *For any strictly decreasing demand function $Q$ and supply $s$ the following holds:*

1. *The minimum admission price is at least $p_{ser}$, and thus transactions paying less than this price will never be included. In particular, the dynamics stay always between $p_{ser}$ and $p_{mon}$, that is, prices $p_t$ satisfy $p_{ser} \leq p_t \leq p_{mon}$ for all $t \geq 1$. Moreover, at each step $t$, the prices either decrease $(p_t < p_{t-1})$ or they jump up to the monopolist price $(p_t = p_{mon})$.*

2. *For every $\delta > \overline{\delta}_{ser}$, the minimum admission price is at most $p_{ser}^{(\delta)}$ defined by (9) which satisfies $p_{ser} < p_{ser}^{(\delta)} < \overline{p}_{ser}$. Moreover, the dynamics pass through the monopolist price $p_{mon}$ infinitely often.*

3. *Every price larger than $p_{ser}$ is an admission price for a sufficiently large $\delta$. That is, for every $p^\star > p_{ser}$, there exists $\delta_{min}(p^\star) < 1$ such that $p^\star$ is an admission price for every $\delta > \delta_{min}(p^\star)$. Moreover, the dynamics pass through the monopolist price $p_{mon}$ infinitely often.*

*Therefore, transactions paying at least $p_{mon}$ are immediately included, and this is tight as there are infinitely steps for which paying less will delay admission to a later step.*

*Proof.* We distinguish the three parts:

1. The bounds on the prices are due to Lemmas 5 and 6. The condition on the price changes is simply the observation that for $p \geq p_{t-1}$ we have $D_t(p) = Q(p)$ and thus the dynamics either take the monopolist price $(p_t = p_{mon})$ or take a smaller price $(p_t < p_{t-1})$.

2. Consider any $p_{ser}^{(\delta)}$ such that $p_{ser}^{(\delta)} < p^\star < \overline{p}_{ser}$. Lemma 9 together with Lemma 11 (Item 3) imply that there exists $\Delta$ such that for every $T$ there exists some $T < t \leq T + \Delta$ with $p_t \leq p^\star$. Lemma 11 (Item 2) states that the conditions in Lemma 10 hold for some $p^\star < p_{ser}$. The latter implies that the dynamics takes the monopolist price infinitely often.

3. The first part follows directly from Lemma 9 and from the fact that $p^\star > p_{ser}$ implies $Q(p^\star) < Q(p_{ser})$ and thus $\delta_{\min}(p^\star) < 1$. The second part follows from Lemma 10 and Item 2 of Lemma 11.

This completes the proof. $\qquad\square$

