# OpenReview forum: "Serial Monopoly on Blockchains with Quasi-patient Users"
_ICML.cc/2024/Workshop/Agentic_Markets — Agentic Markets @ ICML'24 Poster_

### Official Review · Reviewer_qc8m · 2024-06-06
**Review for submission 27**

**Rating:** 8
**Confidence:** 5

**Review:**

**Summary:** The paper studies the problem of setting transaction fee in a blockchain in which block-producers have a monopolistic power over which transactions to include. At each time step, the block producer has full power to set the inclusion price for transactions and also to decide which transactions to include. It turns out that minor changes in user patience, i.e., whether users wait for the transaction to be processed in case that next-block inclusion failed, play a significant role in price dynamics. The paper finds that that minimum admission prices (and their dynamics) are highly sensitive to user preferences leading to oscillations and, thus, creating incentives for strategic behaviour (which is left as a future work).

**Evaluation:** The paper builds upon recent literature and does a good job in extending state-of-the-art studies. It employs a rigorous analysis and offers a rich set of results that nicely compares with existing findings. The main weakness of the paper seems to be the lack of a robustness analysis (see my last comment below), i.e., analyses of variations of the main setting to demonstrate that the findings are not dependent on the (theoretical) abstraction that is employed. Studying incentives is definitely a "weakness", but this is typically a whole paper on its own and I acknowledge how difficult this can be. However, having such an analysis will considerably strengthen the paper (although, as mentioned, not necessary due to its difficulty). In sum, I think that the paper is interesting and will appeal to the audience of the workshop. Thus, I recommend acceptance and offer some comments to the authors below to potentially help with improving it.

**Comments to authors:**
- Top of page 2: the social welfare is not only the total value of the chosen transactions but also the value of the block producers (see the definition in e.g. the [BGR24] paper) - although how to derive/estimate these values, both for users and the block producers, is another, difficult question).
- Maybe an interesting question is to check how the $\delta \in (0,1)$ to model quasi-(im)patient users compares with the simulations presented here: https://ethereum.github.io/abm1559/notebooks/stationary1559.html, in which users stay in the pool but have an increasing cost of waiting, i.e., of having their transaction included in a future block.
- Overall, I think that the paper needs to include more ablations, i.e., different demand function, different s etc to make its results robust for practical purposes. Otherwise, the results are rich and interesting, but currently may seem brittle to changes in the model's assumptions.
- The phase transition observed in the dynamics, which is mentioned in the abstract, is not really presented in the main text (I searched for that word and couldn't find it). Thus, better aligning claims with actual results is another area of improvement.
- Another possible variation is the PBS environment in which builders compete for getting their block selected by the current proposer. Does this setting violate the monopolistic assumption or not?

---

### Official Review · Reviewer_5kmC · 2024-06-14
**Serial Monopoly on Blockchains with Quasi-patient Users - Review**

**Rating:** 6
**Confidence:** 4

**Review:**

This paper investigates the price dynamics in a serial monopoly mechanism for transaction fees in blockchain systems. This setting was previously examined by Nisan et al. for fully patient users who remain in the system indefinitely until their transaction is included in a block, and by Lavi et al. for totally impatient users who leave the system if their transaction is not immediately included.

The authors extend these settings to include quasi-impatient users, characterized by a parameter $\delta \in [0,1]$ representing the fraction of users who stay in the system after their transaction is included in a block. They provide an interesting characterization of the admissible prices, which are the minimum prices a transaction needs to pay to eventually be included in a block.

The authors prove that the admissible price is always higher than the serial price (see Definition~1). They also demonstrate that for sufficiently large δ, any price above the serial price can be the minimum admissible price.

I believe the paper addresses an important and relevant problem in modern blockchain systems and will contribute significantly to the workshop discussion. Therefore, I recommend its acceptance.